# Evaluation of the Use of Methylation as a New Tool for the Diagnostics and Progression of Squamous Intraepithelial Lesions

**DOI:** 10.3390/ijms252211863

**Published:** 2024-11-05

**Authors:** Dominik Pruski, Sonja Millert-Kalińska, Agata Lis, Ewa Pelc, Przemysław Konopelski, Robert Jach, Marcin Przybylski

**Affiliations:** 1Department of Obstetrics and Gynecology, District Public Hospital, Juraszów 7-19, 60-479 Poznan, Poland; millertsonja@gmail.com (S.M.-K.); nicramp@poczta.onet.pl (M.P.); 2Doctoral School, Poznan University of Medical Sciences, Fredry 10, 61-701 Poznan, Poland; 3Imogena Sp.z.o.o., ul. Jeleniogórska 16, 60-179 Poznan, Poland; agata.lis@mineola.pl (A.L.); ewa.pelc@gmail.com (E.P.); przemyslaw.konopelski@mineola.pl (P.K.); 4Medical Diagnostic Laboratory Mineola, ul. Adama Vetulaniego 14, 31-226 Krakow, Poland; 5Division of Gynecologic Endocrinology, Jagiellonian University Medical College, Kopernika 23, 31-501 Krakow, Poland; jach@cm-uj.krakow.pl

**Keywords:** methylation, risk of progression, observation of LSIL and HSIL, fertility-sparing of HSIL, new marker of progression

## Abstract

Vaccination against human papillomavirus (HPV) significantly reduces the incidence of HPV-related lesions worldwide. Considering the increasingly young age of patients in gynecological offices and earlier sexual initiation and potential contact with the HPV virus, doctors need the tools to verify diagnoses. Currently, women plan to pursue motherhood later, so it is necessary to consider whether sexual treatment in the form of, among others, loop electrosurgical excision procedures (LEEPs) may increase the risk of premature birth or difficulty dilating the cervix during labour. For this reason, to avoid the overtreatment of low-grade squamous intraepithelial lesions (LSILs), methylation testing may be considered. In patients with histopathologically confirmed high-grade squamous intraepithelial lesions (HSILs) during biopsy and, ultimately, a lower diagnosis, i.e., LSIL or no signs of atypia, methylation was found to be a useful tool. We performed a Pap smear, HPV genotyping, a punch biopsy, LEEP-conization (if needed), and methylation tests on 108 women admitted to the District Public Hospital in Poland. Women with a negative methylation test result were significantly more likely to be ultimately diagnosed with LSIL (*p* = 0.013). This means that in 85.7% of the patients with HSIL, major cervical surgery could be avoided if the methylation test was negative. Methylation testing, as well as dual-staining and diagnostics detecting the mRNA transcripts of highly oncogenic types of HPV, might be used in the future in the diagnosis of pre-cancerous conditions, mainly of the cervix, and in HPV-dependent cervical cancer screening. The methylation test may also be used in the diagnosis and identification of lesions within the cervical canal, including those located deep within the frontal crypts, not visible even during a professional colposcopic evaluation of the cervix.

## 1. Introduction

According to the newest worldwide data from GLOBOCAN 2022, there are 661,021 new cases of cervical cancer per year and more than half of patients die each year (348,189) [1]. It is the most common cancer among women after breast, lung, and colon cancer, and the most lethal, which is surprising in the 21st century and in the era of modern diagnostics and prevention. It is the only cancer against which primary prevention is available in the form of vaccination. The introduction of vaccines against HPV has revolutionised cervical cancer prevention [2]. Vaccinations against HPV are included in the primary prevention of precancerous lesions—mainly squamous intraepithelial lesions (SIL) and cervical cancer. The other cancers associated with HPV infections affect the genital organs (vulva, vagina, and penis), anal canal, oral cavity, and upper respiratory tract [3,4]. Vaccination against HPV has significantly reduced the incidences of HPV-related lesions in New Zealand and the United States. Currently, there are three HPV prophylactic vaccines available commercially in Europe: Gardasil^®^4 (quadrivalent vaccine against HPV16, 18, 6, and 11, available since 2006), Cervarix™ (bivalent vaccine against HPV16 and 18, approved by EMA in 2007 and the FDA in 2009), and Gardasil^®^9 (nonavalent vaccine against HPV6, 11, 16, 18, 31, 33, 45, 52, and 58, available since 2014) [5]. Data from clinical trials have proved their safety and preventive effect in people infected with HPV. Currently, in Poland, there are two vaccines—bivalent and nonavalent.

An increasing awareness contributes to the fact that an increasing percentage of women visiting gynecological offices are already vaccinated—either in childhood or adulthood. This is related to the growing number of scientific reports supporting the benefits of vaccination in people after sexual initiation and during the treatment of SIL [6]. This determines the differences in the distribution of detected HPV genotypes in cervical smears and necessitates new cervical triage/screening methods.

The vast majority of cervical cancers are high-risk HPV (hrHPV)-related, and the implication of this virus in cervical cancer pathobiology is well known—namely, its effect on the transformation of epithelial surfaces such as the squamous–columnar junction of the cervix [7]. Neoplastic transformation begins with HPV DNA integrating into the genome of a normal epithelial cell.

Changing behaviour, earlier initiation, and the presence of some vaccinated patients have caused the profiles of women with cervical intraepithelial lesions to change. With the lower age of patients, doctors’ approach to the therapeutic process may change from less invasive to more conservative. This is related to decisions regarding pursuing motherhood later and the potential impact of cervical excision procedures, including LEEP-conization, on difficulties in maintaining pregnancy due to cervical scarring. Moreover, some patients are involved in the therapeutic process and want more conservative treatment; these patients seek methods to monitor early changes in their intraepithelial neoplasia, such as LSIL or HSIL, which are important.

Such methods include methylation or CINtec testing based on dual-staining or mRNA transcript detection. The CINtec PLUS test is an immuno-cytochemical test that examines the expression of the p16 protein and Ki-67 protein in cervical epithelial cells. The level of the p16 protein increases in cells infected with the HPV virus, which is the main factor leading to cervical cancer. In turn, the amount of the Ki-67 protein increases in rapidly dividing cells, such as cancer cells. Testing both markers simultaneously increases the sensitivity of the test and allows for the differentiation of changes occurring in cervical cells, detecting cancerous changes caused by the highly oncogenic human papillomavirus at an early stage of cancer development. DNA methylation, one of the most extensively studied epigenetic mechanisms regulating gene expression, has emerged as a promising source for non-invasive disease biomarkers. In the context of cervical cancer, alterations in the promoter methylation levels of various genes—both human and HPV-related—have been linked to HPV status, lesion progression, and patient outcomes [8,9,10]. Although the sensitivity and specificity of the three methods—methylation, mRNA detection, and CINtec—are high, they differ in their capture points. The main targets of methylation tests are epigenetic changes; in CINtec, it is an overexpression of p16 and Ki-67; and in the case of mRNA transcript detection tests, it is the expression of viral oncogenes (E6/E7 mRNA).

There is evidence that type-specific DNA methylation shows a good risk stratification for women with HPV and improved performance compared to HPV16/18/31/45 genotyping and Pap smears, suggesting that HPV DNA methylation has the potential for clinical use as a triage test for HPV-positive women. Our study aims to demonstrate that the methylation test applies to patients with histopathologically confirmed squamous intraepithelial neoplasia of the cervix.

## 2. Results

The analysis included 108 women with an average age of 40 years, of whom 85 were included in the study group and 23 in the control group. The control group consisted of patients whose histopathological diagnosis showed no pathology. Table 1 shows the characteristics of the patients divided into the study and control groups. The methylation outcome (positive/negative) was significantly different between the groups, *p* = 0.004. The research group was characterized by a higher proportion of positive outcomes (43.5% vs. 8.7%, n = 2) and a lower proportion of negative outcomes (56.5% vs. 91.3%).

The HPV outcome (positive/negative) was significantly different between the groups, *p* < 0.001. The research group was characterized by a higher proportion of positive HPV outcomes (85.9% vs. 34.8%, n = 8) and a lower proportion of negative HPV outcomes (14.1% vs. 65.2%). Analysis of HPV genotypes confirmed that the incidences of genotype 16 and genotype 31 differed significantly between the groups, *p* = 0.011 and *p* = 0.038, respectively.

### 2.1. Comparison Between Patients with a Positive (+) and Negative Outcome of Methylation (−) in the Research Group

The Pap smear outcome significantly differed between groups, *p* = 0.016. ASC-H and HSIL were more common among patients with positive methylation outcomes (32.4% vs. 20.8%, 29.7% vs. 6.2%, n = 3, respectively). NILM, ASC-US, and LSIL were less commonly observed among patients with positive methylation outcomes (8.1%, n = 3 vs. 12.5%, n = 6, 8.1%, n = 3 vs. 25.0% and 16.2%, n = 6 vs. 31.2%). HPV genotype 16 was more common among patients with positive methylation outcomes (59.5% vs. 33.3%), *p* = 0.029.

The groups differed in LEEP/hysterectomy outcomes in a significant way, *p* = 0.019. HSIL was more common among patients with positive methylation outcomes (75.7% vs. 58.3%). LSIL was less commonly observed among patients with positive methylation outcomes (8.1%, n = 3 vs. 25.0%). Adenocarcinoma and clinically suspicious outcomes were only observed among patients with a positive outcome of methylation, both with a proportion of 5.4% (n = 2). Patients with no signs of atypia were less common in the group with positive methylation outcomes (2.7%, n = 1 vs. 12.5%, n = 6).

Outcome significantly differed between groups, *p* = 0.048. LSIL was less common among patients with positive methylation outcomes (10.8%, n = 4 vs. 22.9%). The proportion of HSIL was similar in both groups (78.4% vs. 77.1%). Adenocarcinoma and clinically suspicious images of the cervix were only observed among patients with a positive outcome of methylation, both with a proportion of 5.4% (n = 2), as presented in Table 2.

### 2.2. Effectiveness of Methylation in Predicting Final Outcome in the Research Group

The sensitivity of methylation when predicting the final outcome was 47.14% (CI95%: 35.09–59.45%), while the specificity was 73.33% (CI95%: 44.90–92.21%), as shown in Table 3.

### 2.3. Effectiveness of Methylation in Predicting Final Outcome in Both Study Groups (Research Group + Control Group)

The sensitivity of methylation when predicting the final outcome was 47.14% (CI95%: 35.09–59.45%), while the specificity was 84.62% (CI95%: 69.47–94.14%), as presented in Table 4.

Patients with no biopsy result or no LEEP-conization result were excluded from the following analysis, i.e., only patients with complete histopathological results were included. Among patients with HSIL diagnosed in biopsy and LSIL confirmed during the LEEP-conization procedure, statistically significantly more patients had a negative methylation result than a positive one (*p* = 0.013). The sensitivity of the methylation results in predicting a lower final diagnosis for a biopsy result of HSIL was 91.67%, as shown in Table 5 and Table 6.

In the case of patients with a histopathological diagnosis of LSIL in a biopsy, in whom the final diagnosis was confirmed, the methylation result was negative in 85.7% and positive in 14.3%, but no statistically significant results were obtained (*p* = 0.119), as presented in Table 7 and Table 8.

## 3. Discussion

Our study aimed to evaluate the methylation test as a modern predictive tool for use in patients with histopathologically confirmed cervical squamous intraepithelial neoplasia. Considering the increasingly young age of patients in the gynecological office, along with earlier sexual initiation and potential contact with the HPV virus, doctors must have the tools to verify diagnoses. Currently, women plan for motherhood later, so it is necessary to consider whether sexual treatment in the form of, among others, LEEP-conization may increase the risk of premature birth or difficulty dilating the cervix during labour [11,12].

For this reason, and to avoid the overtreatment of LSIL changes, methylation testing may be considered. In patients with histopathologically confirmed HSIL during biopsy and ultimately a lower diagnosis—i.e., LSIL or no signs of atypia—methylation was found to be a useful tool. Women with a negative methylation test result were significantly more likely to be ultimately diagnosed with LSIL (*p* = 0.013). This means that, in 85.7% of patients with HSIL, major cervical surgery could be avoided if methylation was negative. However, no significant results were obtained in the case of LSIL biopsy results. This means that methylation is not yet a tool for monitoring CIN1 lesions, even though the discrepancy in the results seemed to be large. It would probably be worth repeating this study on a larger group of patients. The high percentage of results consistent with both targeted biopsy and LEEP-conization proves the high quality of colposcopy performed in our centre, without overdiagnosis and overtreatment, but also without underestimation.

Salta S. and co-authors published the results of a meta-analysis regarding DNA methylation as a triage marker for colposcopy referral in HPV-based cervical cancer screening [13]. They indicate an accurate test for identifying clinically relevant hrHPV infections is key to reducing the number of unneeded referrals and interventions (with associated risks and costs) as well as hrHPV test repetitions [14,15]. The sensitivity and specificity for CIN2+ detection were 68% (CI 95% 63–72%) and 75% (CI 95% 71–80%), respectively. In our study, the sensitivity and specificity of methylation in predicting the LSIL outcome in LEEP-conization achieved 91.67% and 52.67%, respectively. In a 2024 paper, Hoyer’s team of researchers and co-authors confirmed the possibility of using methylation to detect the recurrence of HSIL—in CIN2 and CIN3 lesions. The sensitivity of the methylation test was 67%, compared to the hrHPV test at 83%, while the specificity was 90% and 62%, respectively, in favour of the test GynTect methylation; these results were statistically significant. The authors also confirmed the higher diagnostic value of the combined methylation test and cytology test compared to the combined hrHPV molecular test and cytology test. However, it is worth noting a significant limitation of this study; the size of the study group was only 17 cases [16]. Methylation tests can also be a complementary diagnosis in the case of persistent infection with HPV types 16, 18, and 59. In the study by Peronace et al., the percentage of positive methylation results increased with more abnormal cytological test results. For the diagnosis of HSIL, it was over 84% [17]. In 2023, Li et al. conducted research on a large group of 476 female patients. *PAX1m* was significantly increased in HSIL, especially in cervical cancer, but there was no significant difference between the cervical intraepithelial neoplasms CIN1 and CIN2. However, HPV VL significantly differed between CIN1 and CIN2, but not between CIN3 and cervical cancer [18]. Most publications confirm the need to use methylation based on co-testing or as an additional diagnostic measure in the event of abnormal screening results. In a 2019 study by del Pino et al., the high sensitivity and specificity of the methylation test was confirmed, which were 84.6% and 74%, respectively, in detecting HSIL changes in CIN2 and CIN3 and cervical cancer. The combined test with molecular diagnostics had an even higher sensitivity and specificity of 80.7% and 85.1%, respectively. The methylation rate of CADM1, MAL, and miR124 increases with the severity of the lesion [19]. In recent years, DNA methylation-based biomarkers have been explored as potential tools for triaging hrHPV-positive cases, aiming to reduce the number of referrals for colposcopy and prevent overdiagnosis and overtreatment. However, the evidence supporting these triage tests remains limited, as noted in the latest World Health Organization recommendations [20]. To address this, we conducted a study to assess the value of DNA methylation-based biomarkers in the Polish population.

## 4. Materials and Methods

### 4.1. Study Design

We describe a prospective, ongoing, 24-month non-randomized study in patients reporting to the District Public Hospital in Poznan, Poland, and Individual Specialized Gynecological Practice for an in-depth diagnostic assessment of squamous intraepithelial lesions. The study group included patients referred for in-depth diagnostics, i.e., colposcopy with cervical biopsy and curettage of the cervical canal, who were qualified based on meeting at least one of the following criteria: (1) abnormal cytology result; (2) positive test result for the highly oncogenic HPV virus (especially 16, 31, and 18); (3) clinically abnormal image of the cervix. Abnormal LBC results mean either ASC-US (atypical squamous cells of undetermined significance), AGC (atypical glandular cells), LSIL (low-grade squamous intraepithelial lesions), HSIL (high-grade squamous intraepithelial lesions), or suspicion of cervical cancer. A cytodiagnostic test was performed on the entire group of patients, i.e., a cervical smear indicating the presence of methylation. LEEP-conization was performed in cases of histopathologically confirmed HSIL in cervical biopsy. In some cases, e.g., clinical suspicion of a highly advanced lesion, the cervical biopsy stage was omitted and LEEP-conization was performed. In some clinical situations, a hysterectomy was performed instead of LEEP-conization of the cervix. For the final histopathological diagnosis, the highest diagnosis from biopsy, LEEP-conization or hysterectomy was considered. Patients whose final histopathological results were negative for squamous intraepithelial lesions were included in the control group.

### 4.2. Specimen Collection and Handling

#### 4.2.1. HPV Genotyping Test and LBC

The BD Onclarity HPV Assay is a method for detecting 14 different HPV genotypes while incorporating a β-globin internal control (IC) as a processing control. The HPV genotypes were detected using specific primers designed to target a region of 79 to 137 bases in the E6/E7 genome, while the IC primers target a 75-base region in the human β-globin gene. To perform the assay, three polymerase chain reaction (PCR) assay tubes—G1, G2, and G3—were utilized, along with four optical channels for the detection process. The HPV genotypes that can be detected individually are HPV16, 18, 31, 45, 51, and 52. The remaining genotypes are grouped as follows: P1—HPV33/58, P2—56/59/66, and P3—35/39/68. The IC served as a control to ensure the accuracy and validity of the assay.

#### 4.2.2. Sample Preparation for Methylation Test

Isolation from cytological cervical material

Genomic DNA was extracted from the Cell Collection Medium 20 mL (Roche Diagnostics, Bazel, Switzerland) using the TANBead OptiPure Viral DNA/RNA Kit based on magnetic bead technology. The extraction was carried out on the TANBead Maelstrom™ 4800 Nucleic Acid Extraction System from Taiwan Advanced Nanotech Inc. (Taoyuan City, Taiwan). Sample preparation for the extraction stage was as follows: 3 mL of the sample from the Cell Collection Medium was centrifuged for 5 min at 7000 r.p.m. to obtain a cell pellet. The supernatant was removed to avoid damaging the pellet at the bottom of the test tube. Then, the sediment was washed with Tris buffer (SIGMA TRIS SALINE, pH 8.0, Merck KGaA, Darmstadt, Germany) and centrifugated for 5 min at 7000 r.p.m. The supernatant was removed, and the pellet was suspended with 300 µL or 400 µL lysis buffer (GeneMAP Extraction Buffer, GENMARK SAGLIK URUNLERI, Halil Rıfat Paşa Mahallesi Güler Sokak, Okmeydanı, Istanbul), depending on the number of pellets, and 10 µL Proteinase K. Next, the sample was incubated for 90 min at a temperature of 56 °C. After incubation, the material was used directly for the isolation step on the TANBead Maelstrom™ 4800 Nucleic Acid Extraction System (Taiwan Advanced Nanotech Inc., Taoyuan City, Taiwan). The DNA concentration of each sample was measured using a NanoReady Touch Series Micro Volume (UV–Vis) Spectrophotometer from Lifereal Biotechnology Co., Ltd. (Hangzhou city, Zhejiang, China).

B.DNA bisulfite conversion and methylation

DNA samples were subjected to DNA denaturation and bisulfite conversion using a sample pretreatment kit for the DNA methylation test. The methylation step was performed by a DNA methylation detection kit for the human *PAX1, SOX1,* and *HAS1* genes (real-time PCR), Yaneng BIOscience (Shenzhen, China) Co., Ltd. The sample pretreatment kit for the DNA methylation test procedure comprises the following steps:Denaturation and bisulfite conversion;Binding of the DNA to the spin column and desulfonation;Washing the spin column bound DNA and elimination of ethanol;Elution of converted DNA.

The methylation kit is based on a multiplex real-time methylation-specific assay for the detection of promoter hypermethylation of the genes *PAX1*, *SOX1*, and *HAS1* in human cervical specimens. After bisulfite treatment, the primes and probes can distinguish between methylated and non-methylated sequences. The Tagman probes can correspond to the sequences that detect methylation in bisulfite-treated DNA samples. ACTB was used as a reference gene for bisulfite treatment and DNA input. Each run included a positive control and negative control for the operation process and environmental pollution monitoring, respectively. The limit of detection (LOD) of the DNA methylation detection kit for human *PAX1, SOX1, and HAS1* is 1% for each gene. Real-time PCR was performed using a Gentier96R Real-time PCR System (Xian Tianlong Science and Technology Co., Ltd., Shanglin Road, Weiyang District, Xian, Shaanxi, China).The analysis was conducted according to the manufacturer’s manual.

### 4.3. Colposcopy, Punch Biopsy and LEEP-Conisation

Further validation of abnormal screening results was performed on all patients with an abnormal smear above the ASCUS (as follows: ASC-US, AGC, LSIL, HSIL, cervical cancer); a positive HPV test for types 16, 18, and 31; and a clinically suspicious cervical image. The Polish Society of Colposcopy and Cervical Pathophysiology recommends the International Federation of Cervical Pathology and Colposcopy classification. The colposcopic assessment was performed by two independent colposcopists, including one gynecological oncologist with 15 years of experience in optical stereoscopic colposcopy.

### 4.4. Statistical Analysis

Analysis was conducted with statistical software R, version R4.1.2. All analyses assumed a significance level of α = 0.05. Nominal variables were presented as n and %; age was presented as the median with Quartiles 1 and 3 due to non-normal distribution. The normality of the distribution was analyzed using Shapiro–Wilk’s test and further verified for skewness and kurtosis. Comparisons were made using the Mann–Whitney U test, Pearson’s chi-square test, and Fisher’s exact test, as appropriate.

## 5. Conclusions

Methylation testing, as well as dual-staining and diagnostics detecting mRNA transcripts of highly oncogenic types of HPV, might be used in future for the diagnosis of pre-cancerous conditions, mainly of the cervix, and in HPV-dependent cervical cancer screening. The methylation test may also be used in the diagnosis and identification of lesions within the cervical canal, including those located deep within the frontal crypts, not visible even during a professional colposcopic evaluation of the cervix.

## Figures and Tables

**Table 1 ijms-25-11863-t001:** Study group characteristics and comparison of groups.

Characteristics	Research Group	Control Group	MD (95% CI)	*p*
N	85 (100.0)	23 (100.0)	-	-
Age, years, Me (Q1; Q3)	37.97 (33.79;41.92)	49.47 (41.87;58.52)	−11.50 (−16.39;−5.82)	<0.001
Methylation, n (%)				
Positive	37 (43.5)	2 (8.7)	-	0.004
Negative	48 (56.5)	21 (91.3)		
Pap smear, n (%)				
NILM	9 (10.6)	13 (56.5)	-	<0.001
ASC-US	15 (17.6)	4 (17.4)		
LSIL	21 (24.7)	3 (13.0)		
ASC-H	22 (25.9)	1 (4.3)		
HSIL	14 (16.5)	0 (0.0)		
AGC	2 (2.4)	1 (4.3)		
Clinically suspicious image of the cervix	2 (2.4)	1 (4.3)		
HPV, n (%)				
Positive	73 (85.9)	8 (34.8)	-	<0.001
Negative	12 (14.1)	15 (65.2)		
HPV genotype, n (%) *				
HR	34 (40.0)	6 (26.1)	-	0.326
16	38 (44.7)	3 (13.0)	-	0.011
18	4 (4.7)	0 (0.0)	-	0.576
31	14 (16.5)	0 (0.0)	-	0.038
45	5 (5.9)	0 (0.0)	-	0.582
Biopsy, n (%)				
LSIL	16 (18.8)	0 (0.0)	-	<0.001
HSIL	58 (68.2)	0 (0.0)		
Adenocarcinoma	2 (2.4)	0 (0.0)		
Squamous cell carcinoma	2 (2.4)	0 (0.0)		
No signs of atypia	3 (3.5)	13 (56.5)		
Not performed	4 (4.7)	10 (43.5)		
LEEP/hysterectomy, n (%)				
LSIL	15 (17.6)	0 (0.0)	-	<0.001
HSIL	56 (65.9)	0 (0.0)		
Adenocarcinoma	2 (2.4)	0 (0.0)		
Squamous cell carcinoma	2 (2.4)	0 (0.0)		
No signs of atypia	7 (8.2)	5 (21.7)		
Not performed	3 (3.5)	18 (78.3)		
Final, n (%)				
LSIL	15 (17.6)	0 (0.0)	-	<0.001
HSIL	66 (77.6)	0 (0.0)		
Adenocarcinoma	2 (2.4)	0 (0.0)		
Squamous cell carcinoma	2 (2.4)	0 (0.0)		
No signs of atypia	0 (0.0)	23 (100.0)		

Comparisons were made with the Mann–Whitney U test (age), Pearson’s chi-square test (methylation, HPV positive/negative, HPV HR, HPV16) or Fisher’s exact test (other characteristics). * Sum exceeded 100% as patients could belong to multiple groups. Me—median; Q1—first quartile; Q3—third quartile; MD—median difference (research group vs. control group); N—number, LSIL—low-grade squamous intraepithelial lesion; HSIL—high-grade squamous intraepithelial lesion.

**Table 2 ijms-25-11863-t002:** Comparison of patients with positive methylation outcome (+) and negative methylation outcome (−) in both study groups. Comparisons were made with Pearson’s chi-square test (HPV-negative, HPV16, HPV31 in research group) or Fisher’s exact test (other characteristics). * Sum exceeded 100% as patients could belong to multiple groups.

Characteristics	Research Group			Control Group		
	Methylation		*p*	Methylation		*p*
	+	−		+	−	
Pap smear, n (%)						
NILM	3 (8.1)	6 (12.5)	0.016	1 (50.0)	12 (57.1)	0.332
ASC-US	3 (8.1)	12 (25.0)		0 (0.0)	4 (19.0)	
LSIL	6 (16.2)	15 (31.2)		0 (0.0)	3 (14.3)	
ASC-H	12 (32.4)	10 (20.8)		1 (50.0)	0 (0.0)	
HSIL	11 (29.7)	3 (6.2)		0 (0.0)	0 (0.0)	
AGC	1 (2.7)	1 (2.1)		0 (0.0)	1 (4.8)	
Clinically suspicious image of the cervix	1 (2.7)	1 (2.1)		0 (0.0)	1 (4.8)	
HPV negative, n (%)	3 (8.1)	9 (18.8)	0.279	1 (50.0)	14 (66.7)	>0.999
HPV genotype, n (%) *						
16	22 (59.5)	16 (33.3)	0.029	0 (0.0)	3 (14.3)	>0.999
18	2 (5.4)	2 (4.2)	>0.999	0 (0.0)	0 (0.0)	-
31	5 (13.5)	9 (18.8)	0.726	0 (0.0)	0 (0.0)	-
Biopsy, n (%)						
LSIL	5 (13.5)	11 (22.9)	0.208	0 (0.0)	0 (0.0)	>0.999
HSIL	25 (67.6)	33 (68.8)		0 (0.0)	0 (0.0)	
Adenocarcinoma	2 (5.4)	0 (0.0)		0 (0.0)	0 (0.0)	
Squamous cell carcinoma	2 (5.4)	0 (0.0)		0 (0.0)	0 (0.0)	
No signs of atypia	2 (5.4)	1 (2.1)		1 (50.0)	12 (57.1)	
Not performed	1 (2.7)	3 (6.2)		1 (50.0)	9 (42.9)	
LEEP/hysterectomy, n (%)						
LSIL	3 (8.1)	12 (25.0)	0.019	0 (0.0)	0 (0.0)	0.395
HSIL	28 (75.7)	28 (58.3)		0 (0.0)	0 (0.0)	
Adenocarcinoma	2 (5.4)	0 (0.0)		0 (0.0)	0 (0.0)	
Squamous cell carcinoma	2 (5.4)	0 (0.0)		0 (0.0)	0 (0.0)	
No signs of atypia	1 (2.7)	6 (12.5)		1 (50.0)	4 (19.0)	
Not performed	1 (2.7)	2 (4.2)		1 (50.0)	17 (81.0)	
Final, n (%)						
LSIL	4 (10.8)	11 (22.9)	0.048	0 (0.0)	0 (0.0)	-
HSIL	29 (78.4)	37 (77.1)		0 (0.0)	0 (0.0)	
Adenocarcinoma	2 (5.4)	0 (0.0)		0 (0.0)	0 (0.0)	
Squamous cell carcinoma	2 (5.4)	0 (0.0)		0 (0.0)	0 (0.0)	
No signs of atypia	0 (0.0)	0 (0.0)		2 (100.0)	21 (100.0)	

Me—median; Q1—first quartile; Q3—third quartile; MD—median difference (research group vs. control group); N—number; NILM—negative for intraepithelial lesion or malignancy; ASC-US—atypical squamous cells of undetermined significance; LSIL—low-grade squamous intraepithelial lesion; ASC-H—Atypical squamous cells (cannot exclude high-grade squamous intraepithelial lesion); HSIL—high-grade squamous intraepithelial lesion; AGC—atypical glandular cells.

**Table 3 ijms-25-11863-t003:** Sensitivity and specificity of methylation in predicting the final outcome in the research group.

		Final Outcome		Sensitivity, %	Specificity, %	PPV, %	NPV, %	Accuracy, %
		HSIL/Adenocarcinoma/Squamous Cell Carcinoma	LSIL					
Methylation	+	33	4	47.14 (35.09–59.45)	73.33 (44.90–92.21)	89.19 (77.47–95.19)	22.92 (16.94–30.24)	51.76 (40.66–62.74)
	−	37	11					

PPV—positive predictive value; NPV—negative predictive value; LSIL—low-grade squamous intraepithelial lesion; HSIL—high-grade squamous intraepithelial lesion.

**Table 4 ijms-25-11863-t004:** Sensitivity and specificity of methylation in predicting the final outcome in both study groups (research group + control group).

		Final Outcome		Sensitivity, %	Specificity, %	PPV, %	NPV, %	Accuracy, %
		HSIL/Adenocarcinoma/Squamous Cell Carcinoma	LSIL/No Signs of Atypia					
Methylation	+	33	6	47.14 (35.09–59.45)	84.62 (69.47–94.14)	84.62 (71.67–92.28)	47.14 (40.78–53.60)	60.55 (50.73–69.78)
	−	37	33					

PPV—positive predictive value; NPV—negative predictive value; LSIL—low-grade squamous intraepithelial lesion; HSIL—high-grade squamous intraepithelial lesion.

**Table 5 ijms-25-11863-t005:** Comparison of patients with HSIL → LSIL/no signs of atypia and other patients. Comparisons made with Pearson’s chi-square test.

Characteristics	Biopsy = HSIL and LEEP = LSIL/No Signs of Atypia n = 12	Other Patients n = 67	*p*
Methylation, n (%)			
Negative	11 (91.7)	32 (47.8)	0.013
Positive	1 (8.3)	35 (52.2)	

PPV—positive predictive value; NPV—negative predictive value; LSIL—low-grade squamous intraepithelial lesion; HSIL—high-grade squamous intraepithelial lesion.

**Table 6 ijms-25-11863-t006:** Sensitivity and specificity of methylation in predicting the biopsy/LEEP outcome (Biopsy = HSIL and LEEP = LSIL/No Signs of Atypia).

		Biopsy = HSIL and LEEP = LSIL/No Signs of Atypia	Other Patients	Sensitivity, %	Specificity, %	PPV, %	NPV, %	Accuracy, %
Methylation	−	11	32	91.67 (61.52–99.79)	52.24 (39.67–64.60)	25.58 (20.25–31.76)	97.22 (84.09–99.57)	58.23 (46.59–69.23)
	−	1	35					

PPV—positive predictive value; NPV—negative predictive value; LSIL—low-grade squamous intraepithelial lesion; HSIL—high-grade squamous intraepithelial lesion.

**Table 7 ijms-25-11863-t007:** Comparison of patients with LSIL → LSIL/no signs of atypia and other patients. Comparisons made with Fisher’s exact test.

Characteristics	Biopsy = LSIL and LEEP = LSIL/No Signs of Atypia n = 7	Other Patients n = 72	*p*
Methylation, n (%)			
Negative	6 (85.7)	37 (51.4)	0.119
Positive	1 (14.3)	35 (48.6)	

PPV—positive predictive value; NPV—negative predictive value; LSIL—low-grade squamous intraepithelial lesion; HSIL—high-grade squamous intraepithelial lesion.

**Table 8 ijms-25-11863-t008:** Sensitivity and specificity of methylation in predicting biopsy/LEEP outcome (Biopsy = LSIL and LEEP = LSIL/No Signs of Atypia).

		Biopsy = LSIL and LEEP = LSIL/No Signs of Atypia	Other Patients	Sensitivity, %	Specificity, %	PPV, %	NPV, %	Accuracy, %
Methylation	−	6	37	85.71 (42.13–99.64)	48.61 (36.65–60.69)	13.95 (10.01–19.12)	97.22 (84.88–99.54)	51.90 (40.36–63.29)

PPV—positive predictive value; NPV—negative predictive value; LSIL—low-grade squamous intraepithelial lesion; HSIL—high-grade squamous intraepithelial lesion.

## Data Availability

All raw data are available at corresponding author.

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
