# Peer review of "Evaluation of the Use of Methylation as a New Tool for the Diagnostics and Progression of Squamous Intraepithelial Lesions"

_ijms, 2024, doi:10.3390/ijms252211863_

Round 1
Reviewer 1 Report
Comments and Suggestions for Authors
Marcin Przybylski et al. in “Evaluation of the use of methylation as a new tool for the diagnostics and progression of squamous intraepithelial lesions” show in extremely depth how the methylation test may also be used in the diagnosis and identification of lesions within the cervical canal, in-cluding those located deep within the frontal crypts, not visible even during a professional colposcopic evaluation of the cervix, as well as within the crypts of the palatine tonsils - in diagnostics of head and neck diseases.
I consider original the proposal of DNA methylation-based biomarkers have been explored 226 as potential tools for triaging hrHPV-positive cases, aiming to reduce the number of re- 227 ferrals for colposcopy and prevent overdiagnosis and overtreatment.
The references are appropriate and recent. They support the conceptualizations present in the article.
To improve paragraph 1, authors should add differences beetwen methods including methylation , CINtec testing based on dual-staining and mRNA transcripts detection (Line 75).
Comments on the Quality of English LanguageMinor editing of English language required.
Reviewer 2 Report
Comments and Suggestions for Authors
Marcin et al. investigated the use of methylation testing to improve the diagnosis of HPV-related cervical lesions. In a study of 108 women, they found that methylation testing could help avoid unnecessary surgeries for low-grade lesions (LSIL). Specifically, 85.7% of women with histopathologically confirmed high-grade lesions (HSIL) could have avoided major cervical surgery if their methylation test was negative. The authors suggest that methylation testing, along with other diagnostic tools, could enhance cervical cancer screening and improve the detection of hard-to-visualize lesions in both the cervix and head and neck.
Comments:
The study presents interesting findings, particularly regarding the potential of methylation testing to reduce unnecessary surgical interventions for patients with cervical lesions. The use of such diagnostics could be highly beneficial in improving patient outcomes and preventing overtreatment.
Concern:
The inclusion of "in diagnostics of head and neck diseases" in the abstract, without further elaboration or mention within the body of the article, raises a concern about consistency. Abstracts should reflect the key focus and findings of the paper, and introducing a topic that isn't discussed in detail in the main text can be misleading to readers. If head and neck diagnostics were not addressed substantively in the article, it would be more appropriate to either remove this reference from the abstract or ensure that it is properly explored within the discussion or results sections.
Comments on the Quality of English LanguageMinor editing of the English language is required.
Reviewer 3 Report
Comments and Suggestions for Authors
The authors suggest that HPV DNA methylation could be used as a triage test for women who test positive for HPV. This is based on their finding of a significant difference in methylation outcomes between the study group (with confirmed squamous intraepithelial neoplasia) and the control group (without pathology) .
The study found that methylation testing could have potentially helped avoid major cervical surgery in a significant proportion of patients (85.7%) who were ultimately diagnosed with low-grade squamous intraepithelial lesions but had negative methylation results. The authors mention that methylation tests could be useful as a complementary diagnostic tool in cases of persistent infection with high-risk HPV types like 16, 18, and 59. The results are valuable and interesting, nevertheless, some issues need to be addressed:
1. There are some abbreviated terms that need to be explained the first time they appear in the text.
2. All the results are interesting, but the form of presentation is not. It would be beneficial for readers to incorporate figures and graphs, there are too many tables in the document.
3. Tables must contain abbreviations in footnotes
4. It would be better to add a conclusion section
5. English should be revised for a native speaker
Comments on the Quality of English LanguageEnglish should be revised for a native speaker
